

# Evaluation potential of PGPR to protect tomato against *Fusarium* wilt and promote plant growth

Rizwana begum Syed Nabi[1,2,*], Raheem Shahzad[3,*], Rupesh Tayade[4], Muhammad Shahid[1], Adil Hussain[5], Muhammad Waqas Ali[6] and Byung-Wook Yun[1]

[1] Laboratory of Plant Functional Genomics, School of Applied Biosciences, Kyungpook National University, Daegu, South Korea
[2] Department of Southern Area Crop Science, National Institute of Crop Science, Rural; Development Administration, Miryang, South Korea
[3] Department of Horticulture, The University of Haripur, Haripur, Pakistan
[4] Laboratory of Plant Breeding, School of Applied Biosciences, Kyungpook National University, Daegu, South Korea
[5] Department of Agriculture,, Abdul Wali Khan University, Mardan, Pakistan
[6] School of Biosciences, University of Birmingham, Edgbaston, Birmingham, United Kingdom
[*] These authors contributed equally to this work.

Corresponding author
Byung-Wook Yun, bwyun@knu.ac.kr

## ABSTRACT

Soilborne fungal diseases are most common among vegetable crops and have major implications for crop yield and productivity. Eco-friendly sustainable agriculture practices that can overcome biotic and abiotic stresses are of prime importance. In this study, we evaluated the ability of plant growth-promoting rhizobacterium (PGPR) *Bacillus aryabhattai* strain SRB02 to control the effects of tomato wilt disease caused by *Fusarium oxysporum* f. sp. *lycopersici* (strain KACC40032) and promote plant growth. *In vitro* bioassays showed significant inhibition of fungal growth by SRB02. Inoculation of susceptible and tolerant tomato cultivars in the presence of SRB02 showed significant protection of the cultivar that was susceptible to infection and promotion of plant growth and biomass production in both of the cultivars. Further analysis of SRB02-treated plants revealed a significantly higher production of amino acids following infection by *F. oxysporum*. Analysis of plant defense hormones after inoculation by the pathogen revealed a significantly higher accumulation of salicylic acid (SA), with a concomitant reduction in jasmonic acid (JA). These results indicate that *B. aryabhattai* strain SRB02 reduces the effects of *Fusarium* wilt disease in tomato by modulating endogenous phytohormones and amino acid levels.

## INTRODUCTION

Tomato (*Solanum lycopersicum* L.) is the second most economically important edible vegetable after potato from the *Solanaceae* family and is widely cultivated and consumed around the world (*Hanson & Yang, 2016*). Tomato is used as a model plant for investigating

the genetics and molecular aspects of disease resistance mechanisms. The tomato crop is under threat worldwide owing to biotic and abiotic stresses that have caused significant reductions in yield and productivity. One reason is that tomato is a host for nearly 200 species of plant pathogens, including fungi, bacteria, nematodes, viruses, and others that infect plants at all developmental stages (*Stout, Kurabchew & Leite, 2017*), reducing both yield and quality.

Vascular wilt is one of the most important fungal diseases of tomato and occurs wherever these crops are grown. This disease is caused by the soilborne fungus *Fusarium oxysporum* f. sp. *lycopersici* (*FOL*). Three different pathotypes have been identified so far, which can be further classified into three races, 1, 2, and 3, based on various pathogenicity features during infection in tomato. Being soilborne, it is omnipresent and is very hard to get rid of once introduced into the cropping system. If infection occurs at the nursery or seedling stage, plants simply die back, whereas severe losses can occur if the disease appears in the field after transplantation. The fungus can spread in different ways, such as through the transport of infested soil, irrigation water, infected plants and transplants, and seeds (*Jones et al., 2014*). Infection occurs via the roots, causing serious vascular damage and wilting of the plant that subsequently leads to cell death. In severe infections, more than 80% of crop loss has been reported (*Worku & Sahe, 2018*). Some studies have reported the applicability of protective fungicides as a possible remedy against the different strains of the pathogen. However, the use of chemicals in agriculture has not only raised serious concerns regarding human health and environmental hazards but is also considered responsible for the development of strains that are resistant to these widely used agrochemicals (*Zouari et al., 2016*). Hence, eco-friendly alternates to chemical measures are needed.

Biological control of plant pathogens has been of great interest to researchers. Apart from pathogenic microbes, plants also have symbiotic or mutualistic interactions with a wide range of soilborne microbes, which protect plants from pathogens either directly or by inducing resistant mechanisms (*Pieterse et al., 2014*). These microbes associate with the plant roots and help enhance growth-related attributes by improving the uptake of essential ions and minerals, atmospheric nitrogen fixation, and protection from pathogens (*Lugtenberg & Kamilova, 2009*). These growth-promoting bacteria are mainly isolated from the rhizosphere of the plants. These microbes are commonly known as plant growth-promoting rhizobacteria (PGPR) (*Kloepper, Lifshitz & Zablotovicz, 1989*; *Backer et al., 2018*) and include organisms such as *Pseudomonas* spp. Other microbes are known as plant growth-promoting endophytic bacteria, plant growth-promoting fungi, or biocontrol fungi (BCF), including *Trichoderma* spp. and *Sebacinales* spp. These can play a role in plant growth and can stimulate plant immune systems (*Shoresh, Harman & Mastouri, 2010*; *Singh et al., 2019*). Endophytes are widely dispersed and can be found in diverse environments including the tropics, temperate zone, aquatics, xerophytics and deserts, tundra, geothermal soils, rainforests, mangroves, and coastal forests. They inhabit plant tissues such as endosperms, roots, leaves, stems, flowers, and fruits (*Singh et al., 2017*). Generally, plant growth promotion may occur owing to the regulation of the plant hormonal system, modifications in root architecture, production of siderophores,

solubilization of soil minerals, activation of secondary mechanisms of plant defense, and production of biochemicals (*Pupin & Nahas, 2014*; *Backer et al., 2018*).

PGPRs and endophytes have a non-pathogenic symbiotic life cycle associated with their host plant tissues; these endophytes can be easily isolated from plant tissues (*Arnold & Lutzoni, 2007*; *Costa et al., 2012*). Seeds are the source of vertical dispersal of numerous seed-borne endophytes, or PGPRs (*Ernst et al., 2003*). Along with the alleviation of biotic stress in plants, these PGPRs have been reported to help mitigate a wide range of abiotic stresses as well (*Shahzad et al., 2017a*). Independent studies have reported the ameliorating effects of PGPRs on plant growth and fungal diseases in tomato and sunflower (*Shittu et al., 2009*; *Waqas et al., 2015*). In addition, studies have revealed the remediation abilities of PGPRs in soil contaminated with heavy metals (*Jing, He & Yang, 2007*; *Bilal et al., 2018*). All of these impacts of PGPRs make them widely attractive as biofertilizers and soil microbe mediators (*Backer et al., 2018*; *Rosier, Medeiros & Bais, 2018*). The positive effects of PGPRs on plant growth attributes are well known, but the exact molecular mechanism(s) behind them have not yet been demonstrated.

PGPRs affect plant growth by either direct or indirect means. The direct promotion of plant growth occurs by a synthesis of complex compounds by the microbes—for instance, phytohormones such as indole-3-acidic acid (IAA), gibberellic acid (GA3), zeatin, and abscisic corrosive (ABA)—or by incremental nutrient accessibility by nitrogen fixation from the surrounding climate, thereby providing supplements for mineral solubilization (*Glick, 1995*; *Bhardwaj et al., 2014*). The indirect method of plant growth promotion takes place when PGPRs get involved in reducing the negative effects of one or more phytopathogenic microbes or fungi. This occurs by the production of substantial antagonistic substances or by inducing resistance in plants against the pathogens; for instance, the production of siderophores, hydrogen cyanide (HCN), hydrolytic proteins, etc. (*Glick, 1995*; *Mahmood, Gupta & Kaiser, 2009*).

The role of antifungal PGPRs as biological control agents to control plant diseases has been widely examined. PGPRs are considered either extracellular, including the genera *Agrobacterium*, *Arthrobacter*, *Azotobacter*, *Azospirillum*, *Bacillus*, *Burkholderia*, *Chromobacterium*, *Erwinia*, *Flavobacterium*, *Micrococcous*, *Pseudomonas*, and *Serratia*, or intracellular, including the genera *Allorhizobium*, *Bradyrhizobium*, *Mesorhizobium*, and *Rhizobium* (*Martínez-Viveros et al., 2010*; *Gouda et al., 2018*). The fact that rhizospheric bacteria *Bacillus aryabhattai* strain B8W22 was previously identified and isolated from cryotubes used for collecting air samples from the earth stratosphere (*Shivaji et al., 2009*) indicates that these bacteria have cosmic ancestry. Moreover, different strains of the bacterium were isolated from the rhizosphere in South Korea, India, and Tibet (*Pailan et al., 2015*; *Lee et al., 2015*; *Yun et al., 2016*). The plant growth-promoting ability of *B. aryabhattai* was initially reported by (*Lee et al., 2012*), who demonstrated growth promotion in *Xanthium italicum* plants. Similarly, *Ramesh et al. (2014)* reported on *B. aryabhattai* contributions to plant growth by enhancing the mobilization and bio-fortification of zinc in soybean and wheat. More recently, *B. aryabhattai* strain SRB02 has been found to play a role in oxidative and nitrosative stress tolerance and promotion of growth in soybean plants by modulating the production of phytohormones (*Park et*

*al., 2017a*). In addition, *B. aryabhattai* strains also show the ability for the biosynthesis of thermostable alkaline phosphatase, anti-leukemic tumor-inhibiting L-asparaginase enzyme (*Gill et al., 2013*; *Singh et al., 2014*), and degradation of pesticides (*Pailan et al., 2015*). In additions, various species of *Bacillus* have been identified as plant growth promoting bacteria as well as biocontrol agents against various pathogenic fungi (*Compant et al., 2005*; *Shahzad et al., 2017a*). Plant growth promoting rhizosphere bacteria employ a variety of strategies to facilitate plant growth and survival under pathogenic attack by both direct and indirect mechanisms. The most common direct mechanisms are phytohormone production, the acquisition of nutrients, and the control of pathogens through various means, for example, through the synthesis of hydrolytic enzymes, antifungal compounds, lipopeptides, or antibiotics. The indirect mechanisms include protection by triggering specific defense-related pathways, particularly the induction of systemic resistance (ISR) against pathogens and pests (*Khan, Mishra & Nautiyal, 2012*; *Martínez-Hidalgo, García & Pozo, 2015*) and the release of bacterial volatile compounds (*Bernier et al., 2011*). However, many environmental factors influence the biological control potential of PGPR by either predisposing pathogens to microbial antagonism, regulating the growth or production of metabolites by specific antagonists, or modulating disease development and consequently the level of disease suppression achieved.

From our literature survey, it is evident that except for some reports in crops (*Xanthium italicum*, soybean, rice, tomato, and wheat) there is a lack of information about the growth-promoting activity of *B. aryabhattai* and its role in tolerance to biotic and abiotic stress in other plant species (*Viljoen et al., 2019*; *Yoo et al., 2019*). In this study, we evaluated the plant growth-promoting abilities of *B. aryabhattai* SRB02 in tomato cultivars inoculated with phytopathogenic fungus *FOL*.

## MATERIALS AND METHODS

### Growth of PGPR and FOL

*B. aryabhattai* SRB02 was isolated previously from the rhizosphere of a soybean field in the Chungcheong buk-do region of South Korea (*Park et al., 2017b*). Bacteria were cultured on LB agar or in broth (AppliChem, Darmstadt, Germany) media at 28 °C for 24 h. *F. oxysporum* f. sp. *lycopersici* strain KACC 40032 was obtained from the Korean Agricultural Culture Collection (KACC, http://genebank.rda.go.kr/) and grown on potato dextrose agar plates at 28 °C for 7 d. The antifungal activity of *B. aryabhattai* SRB02 against *FOL* was evaluated following the protocol of *Shahzad et al. (2017a)*.

Briefly, a 0.5 $cm^2$ disc of active fungal mycelia of *FOL* was placed at the center of a 90 mm disposable plastic Petri dish (SPL, Korea) containing LB agar (Becton, Dickinson and Company, France). The overnight bacterial culture of *B. aryabhatttai* SRB02 was aseptically streaked around the fungal disc at equal distances in a square pattern. For the untreated control, a fungal disc was placed on LB agar, as mentioned earlier, but instead of *B. aryabhatttai* SRB02, only sterile water was streaked. For comparison, the effects of fungal growth inhibition of organic acids against the pathogen were also evaluated. All of the plates were incubated at 28 °C for 7 d. After the incubation period, the inhibition

zone was measured and the percent inhibition was calculated according to the following formula.

$$\text{Inhibition\%} = \frac{(\text{diameter of fungus on control plate} - \text{diameter of fungus on SRB02 co-cultured plate}) \times 100}{\text{diameter of fungus on control plate}}.$$

### Screening of tomato varieties for resistance to FOL

In the current study, tomato seeds of four Korean cultivars (IT 252842-13 (Cultivar-1), IT 252869-14 (Cultivar-2), IT 260627-16 (Cultivar-3), IT 259462-15 (Cultivar-4) ) were selected for their response to the pathogen. Seeds were sterilized with 2.5% sodium hypochlorite for 10 min and kept on wet paper towels inside Petri plates in an incubator at 25 °C for 5 d. Horticultural soil, distilled water, and pots were autoclaved at 121 °C for 20 min. Uniformly germinated seeds were transferred to separate trays filled with sterilized horticultural soil (Soil and Fertilizer Technology, Korea). After one week, uniformly grown seedlings were transplanted to big pots with the dimensions (LxWxH)-3. $5 \times 3 \times 3$ inches and volume 85–90 g. Plants were allowed to acclimatize for a few days, and the experimental treatments were set up in triplicates, with each replicate containing at least six plants. The fungal spore suspension of *FOL* strain KACC 40032 was prepared according to the protocol described by *Lichtenzveig et al. (2006)*. Control plants were treated with distilled water, and plants were allowed to grow for 5 d. Plants to be treated with the pathogen were inoculated by applying a spore suspension ($10^6$ conidia/mL) to the exposed roots of tomato plants. The roots were then covered with soil. Plants were allowed to grow at relatively high humidity of $80 \pm 2\%$. After 14 d of growth under the conditions mentioned above, the inoculated plants were assessed based on symptomatology (severity of plant wilting) and growth.

### In planta biocontrol assessment

After the screening test, two cultivars (resistant and susceptible, one each) were selected based on disease symptoms and growth under biotic stress. Seeds of the selected cultivars were surface-sterilized, germinated, and grown before being transplanted to pots as mentioned previously. The plants were allowed to acclimatize for a few days, and the experimental treatments were set up in triplicate, with each replicate containing at least six plants. SRB02 was applied to plants by soil drenching with 10 mL SRB02 broth culture ($4 \times 10^8$ cfu/mL) in the root zone. The fungal spore suspension *FOL* strain KACC 40032 was prepared as mentioned previously. Control plants were treated with distilled water, and plants were allowed to grow for 5 d. Plants to be treated with the pathogen were inoculated by applying spore suspension ($10^6$ conidia/mL) to the exposed roots of tomato plants. The roots were then covered with soil. The plants were allowed to grow at relatively high humidity of $80 \pm 2\%$ because to further exploit the pathogenic impact of fungus. Data were recorded on growth parameters such as plant height (PH), root length (RL), fresh weight (FW), dry weight (DW), and chlorophyll content (Chl. Cont.) to determine the response of plants to infection in the presence or absence of SRB02. For fresh plant

biomasses, the plants were uprooted, carefully washed, and frozen in liquid nitrogen, and then transferred to storage at −80 °C until further analysis.

## Extraction and quantification of amino acid content

The plant amino acids were extracted according to the protocol described by *Khan et al. (2017)*, with some modifications. Briefly, the freeze-dried whole plant samples were ground to homogenate, and 100 mg powdered samples were hydrolyzed under a vacuum in 6N HCl at 110 °C followed by 80 °C for 24 h. The dried residue was suspended in 0.02N HCl and filtered through a 0.45 μm filter. The amino acids were then quantified using an automatic amino acid analyzer (Hitachi, Japan; L-8900). The experiments were conducted in triplicate, and each replicate was comprised of six plants. The amino acid concentration was determined using relevant standards. This standard known as amino acid standard mixture solution (type H) used for the automatic amino acid investigation was procured through Wako Pure Chemical Industries Ltd (Japan), and used for endogenous amino acids assessment.

## Jasmonic acid quantification

For the quantification of endogenous jasmonic acid (JA) content, the optimized protocol described by *McCloud & Baldwin (1997)* was used. Briefly, homogenized powder (0.3 g) from the immediately freeze-dried whole plant samples was suspended in extraction buffer (70:30 v/v acetone and 50 mm citric acid), and 25 ng JA internal standard ([9, 10-2H$^2$]-9, 10-dihydro-JA) was also added to the suspension. The extract suspension was kept overnight at room temperature for evaporation of highly volatile organic solvents and to retain the less-volatile fatty acids. The subsequent aqueous phase was filtered and then extracted with 30 mL diethyl ether three times. The collective extracts were subsequently loaded onto a solid-phase extraction cartridge (500 mg of sorbent, aminopropyl). In addition, 7.0 mL of trichloromethane and 2-propanol (2:1 v/v) were used to wash the loaded cartridges. Then, the exogenous JA and relevant standard were eluted with one mL of diethyl ether and acetic acid (98:2 v/v). Following evaporation, the samples were esterified and analyzed by GCMS (6890N network GC system) and a 5973 network mass selective detector (Agilent Technologies, Palo Alto, CA, USA) in the relevant ion mode. The relevant ion mode was selected for JA determination. The ion fragment was examined at $m/z = 83$ AMU, corresponding to the base peaks of JA and [9, 10-2H$^2$]-9, 10-dihydro-JA. The endogenous JA values were determined from the peak areas for relevant standards.

## Salicylic acid (SA) quantification

The SA of SRB02-treated tomato plants was extracted and quantified according to the protocol described by *Enyedi et al. (1992)*, *Seskar, Shulaev & Raskin (1998)*. Immediately freeze-dried whole plant tissues were homogenized, and 0.2 g of homogenate powder was used for the extraction using 90% and 100% methanol. The pellets were dried and re-suspended in 2.5 mL 5% trichloroacetic acid (TCA) and further partitioned with ethyl acetate, cyclopentane, and isopropanol (ratio of 100:99:1, v/v). The upper organic layer containing free SA was used for air-drying with nitrogen gas. The dry SA was again suspended in one mL 70% methanol and subjected to high-performance liquid

chromatography (HPLC) using a Shimadzu device outfitted with a fluorescence indicator (Shimadzu RF-10AxL) with excitation at 305 nm and emission at 365 nm filled with a C18 reverse-phase HPLC column (HP Hypersil ODS, particle size 5 μm, pore size 120 Å, Waters). The flow rate was maintained at 1.0 mL/min.

## Statistical analysis

All experiments were replicated three times, and each replicate was comprised of six plants. Data were statistically evaluated with Duncan multiple range tests and $t$-tests where appropriate, using SAS version 9.2 software (Cary, NC, USA).

## RESULTS

### In vitro antifungal assay

The in vitro antifungal activity of PGPR *B. aryabhattai* SRB02 was assessed against pathogenic *Fusarium oxysporum* in dual culture. The results revealed that the PGPR *B. aryabhattai* SRB02 significantly inhibited the growth of pathogenic *F. oxysporum,* as shown in Fig. S1.

### Response of tomato cultivars under pathogenic infection by *F. oxysporum*

To determine the response of four tomato cultivars, the plants were challenged with a spore suspension of the pathogen. The pathogen was applied to the exposed roots of tomato plants and incubated under higher relative humidity to create a conducive environment for successful infection. After 14 d of inoculation, the cultivars revealed a differential level of tolerance to the pathogen (Fig. 1). The plant tolerance level was determined based on the symptomatology (severity of plant wilting). In the susceptible plants, clear symptoms of wilting were evident. Susceptible plants were also observed with retarded growth as compared to the tolerant plants. Based on the plant growth attributes and resistance level, as shown in Fig. 1, the most tolerant and susceptible tomato cultivars were selected for further experiments.

### Plant growth-promoting and ameliorative effects of *B. aryabhattai* SRB02 against FOL

Based on screening, the plant-growth-promoting and biocontrol efficiency of PGPR *B. aryabhattai* SRB02 against a virulent strain of *F. oxysporum* was investigated in both the selected tolerant and susceptible varieties (Fig. 2). *B. aryabhattai* SRB02 significantly promoted plant growth, and interestingly reduced the disease in both tolerant and susceptible tomato cultivars (Fig. 2).

The growth-related traits of the disease-tolerant plants were significantly improved when applied with SRB02 alone. The plant height (PH) was improved by 37.4%, while RL was improved by 26.8% as compared to the water-treated control plants. Other traits including seedling FW, seedling DW, and chlorophyll content were also improved by 15.3%, 23.3%, and 5.8%, respectively. A similar trend was also observed when the tolerant plants were treated with the pathogen and SRB02 together, compared to the pathogen-treated plants. The PH, RL, seedling FW, DW, and chlorophyll content were improved by 124%, 6.4%,

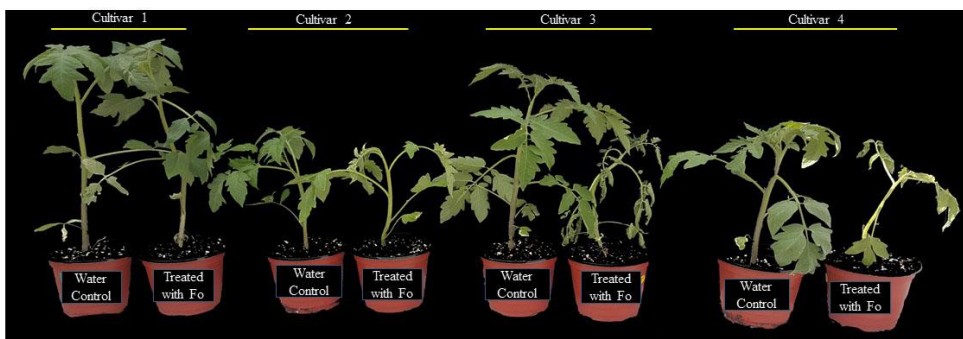

**Figure 1** Response of four tomato cultivars against *F. oxysporum f. sp. Lycopersici*. Cultivar 1 showed tolerance, cultivar 2 was moderately tolerant, and cultivars 3 and 4 were susceptible based upon the symptomatology.

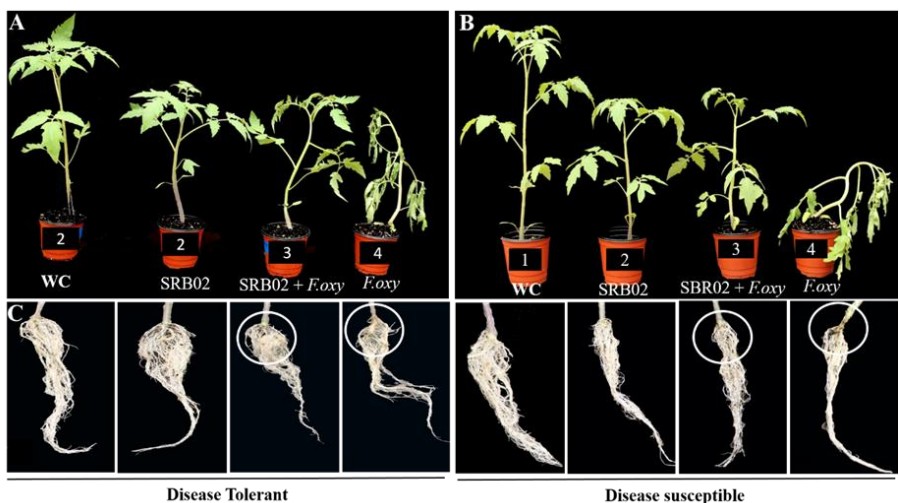

**Figure 2** Effect of SRB02 on susceptible and tolerant tomato plants under inoculation with pathogenic *F. oxysporum f. sp. lycopersici*. The effect of SRB02 on tomato disease-tolerant plants. (A) and disease-susceptible plants (B) to ameliorate disease symptoms and promote growth parameter SRB02 improved the roots of both tolerant and susceptible plants in the presence and absence of the pathogen (C).

15.8%, 42.3%, and 39.7%, respectively, compared to the plants inoculated with the pathogen alone. SRB02 also improved the growth attributes of the disease-susceptible plants with or without co-treatment by the pathogen. The PH of the disease-susceptible plants was improved by 14.1% with the application of SRB02 in comparison with the water-treated control plants; however, the increase in PH was significantly greater (105.7%) in plants treated with SRB02 and *F. oxysporum* combined as compared to the pathogen-treated plants. Likewise, other traits were also improved in plants treated with PGPR alone as compared to the water-treated plants and also in the PGPR and pathogen co-treated susceptible plants in comparison with the plants treated with the pathogen alone. The RL, seedling FW, seedling DW, and chlorophyll content were improved by 9% and 44.5%,

**Table 1** Effects of SRB02 on growth parameters of tomato plants under control and biotic stress conditions.

| Variety | Treatment | PH (cm) | RL (cm) | FW (g) | DW (g) | Chl.Cont. (SPAD) |
|---|---|---|---|---|---|---|
| **Tolerant** | Control | 28.73 ± 0.40b | 15.16 ± 0.34b | 14.38 ± 0.05b | 6.09 ± 0.25b | 28.38 ± 0.37b |
| | SRB02 | 39.48 ± 0.96a | 19.23 ± 0.64a | 16.58 ± 1.00a | 7.51 ± 0.60a | 30.03 ± 0.40a |
| | *F. oxysporum* | 17.50 ± 0.51b | 14.25 ± 0.48b | 13.59 ± 0.01b | 5.18 ± 0.10b | 23.38 ± 0.38b |
| | SRB02 + F. oxy | 39.20 ± 0.88a | 15.17 ± 0.34a | 15.74 ± 0.42a | 7.37 ± 0.27a | 32.67 ± 0.48a |
| **Susceptible** | Control | 29.88 ± 0.45b | 18.75 ± 0.43b | 16.11 ± 0.18b | 6.89 ± 0.49b | 27.11 ± 0.34b |
| | SRB02 | 34.10 ± 0.71a | 20.43 ± 0.58a | 14.42 ± 0.04a | 7.10 ± 0.08a | 28.17 ± 1.42a |
| | *F. oxysporum* | 14.63 ± 0.62b | 13.41 ± 0.76b | 10.83 ± 0.57b | 4.88 ± 0.18b | 25.11 ± 1.07b |
| | SRB02 + *F. oxy* | 30.10 ± 0.53a | 19.38 ± 0.37a | 14.36 ± 0.27a | 6.08 ± 0.08a | 40.51 ± 0.98a |

Notes.

PH, plant height; RL, root length; FW, fresh weight; DW, dry weight; Chl, Cont. chlorophyll content.

Data represent means ± SD of six replicates from three independent experiments. Each values in columns followed by different letters are significantly different at $P \geq 0.05$.

10.4% and 32.6%, 3%, and 24.6%, and 4% and 61.3% in plants treated with SRB02 alone and in plants co-treated with PGPR and the pathogen, respectively (Fig. 2, Table 1).

### *B. aryabhattai* regulates defense against *F. oxysporum* by modulating defense-related hormones in tomato

Measurement of basal and induced levels of the plant defense-related hormones SA and JA following inoculation with *FOL* in the absence or presence of SRB02 revealed strict regulation of plant defense responses in SRB02-treated plants due to the regulation of the synthesis of both of these hormones (Figs. 3 and 4). Interestingly, these results were observed in both the resistant and susceptible cultivars, indicating the high utility of SRB02 for field use even in susceptible crops. More specifically, SRB02-treated infected plants (tolerant and susceptible) produced significantly lower JA (11.10% and 10.30%, respectively) compared to control plants (Fig. 3). Even the SRB02-treated plants in the absence of *FOL* accumulated lower JA (6.92% and 17.91%).

Furthermore, SRB02 treatment with *F. oxysporum*-inoculated plants of the tolerant cultivar accumulated 48.48% more SA compared to plants not treated with the PGPR. More interestingly, the response of the *F. oxysporum*-inoculated plants of the susceptible cultivar was more robust in the presence of SRB02, as these plants produced 74.60% more SA as compared to plants not treated with PGPR (Fig. 4). However, no significant differences in SA accumulation were observed in SRB02-treated plants of either tolerant and susceptible cultivars in the absence of *F. oxysporum*.

### *B. aryabhattai* SRB02 regulates amino acids in plants with or without biotic stress

The current study showed that *B. aryabhattai* SRB02 regulates amino acids in both disease-tolerant and susceptible tomato plants in the presence or absence of *F. oxysporum* (Table 2). Under pathogenic infection by *F. oxysporum*, *B. aryabhattai* SRB02 inoculation significantly enhanced aspartic acid (115.57% and 147.48%), threonine (123.18% and 118.56%), serine (123.13% and 158.91%), glutamic acid (4.86% and 157.89%), glycine (131.82% and 143.58%), alanine (99.61% and 109.67%), valine (98.13% and 74.62%), methionine (239.06% and 172.93%), isoleucine (42.60% and 97.82%), leucine (103.21% and 58.03%),

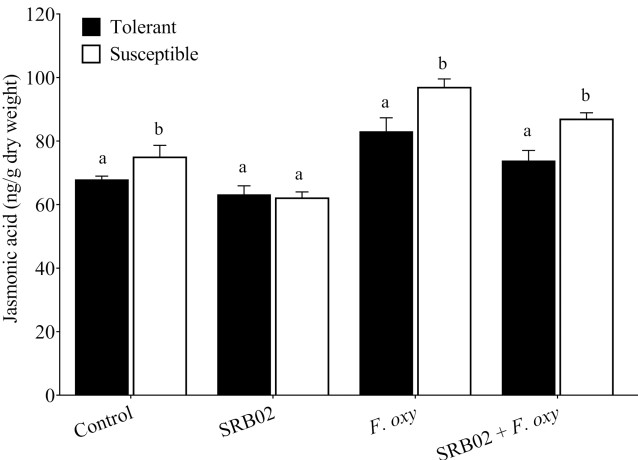

**Figure 3 Regulation of jasmonic acid accumulation in plants.** Under pathogenic stress by *Fusarium oxysporum*, *B. aryabhattai* SRB02 significantly reduced the endogenous JA accumulation level in both susceptible and tolerant plants. Conversely, a reduction in JA accumulation was also observed when SRB02 was applied to the plants in the absence of the pathogen. The significant increase in JA accumulation was observed only in *F. oxysporum*-inoculated plants. Data are means ($\pm$SD) of at least three replications. Means was analyzed for significant differences using Student's $t$-test. Bars with different letters are significantly different at $P < 0.05$ based.

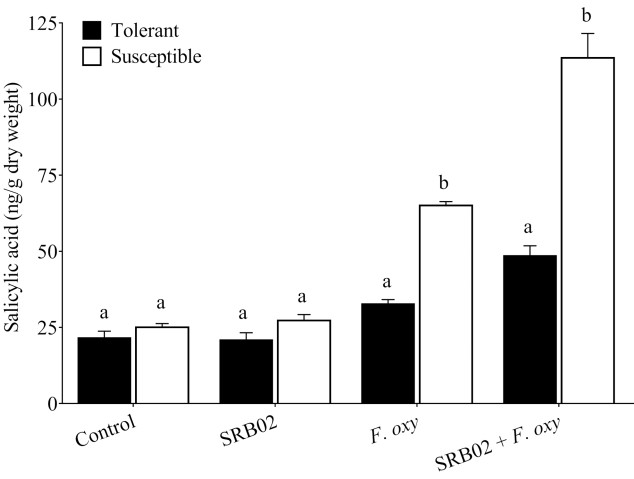

**Figure 4 Regulation of salicylic acid accumulation in plants.** In the presence of the pathogen, *B. aryabhattai* SRB02 significantly enhanced SA accumulation in both tolerant and susceptible plants, while in the absence of the pathogen, the plants displayed no significant changes in SA accumulation under the presence of *B. aryabhattai* SRB02. Data are means ($\pm$SD) of at least three replications. Means were analyzed for significant differences using Student's $t$-test. Bars with different letters are significantly different at $P < 0.05$ based.

tyrosine (138.45% and 65.45%), phenylalanine (39.86% and 34.16%), lysine (113.15% and 98.03%), histidine (98.42% and 111.74%), arginine (108.69% and 157.18%), and proline (90.09% and 115.25%) in disease-susceptible and tolerant tomato plants, respectively (Table 2). Only cysteine was decreased by 9.65% and 21.82% in *B. aryabhattai* SRB02 applied to susceptible and tolerant plants, respectively (Table 2).

Likewise, in the absence of the pathogen, *B. aryabhattai* SRB02 significantly enhanced aspartic acid (3.35% and 24.98%), threonine (32.99% and 118.56%), glutamic acid (4.86% and 157.89%), glycine (30.78% and 143.58%), alanine (29.70% and 4.65%), cysteine (44.22% and 60.20%), valine (21.91% and 31.81%), methionine (132.35% and 31.17%), isoleucine (97.76% and 29.48%), leucine (36.52% and 32.40%), phenylalanine (114.92% and 77.20%), histidine (22.81% and 41.48%), and arginine (35.98% and 23.95%) in the susceptible and tolerant plants, respectively (Table 2). However, *B. aryabhattai* SRB02 showed an increase in serine-intolerant (9.73%) plants and a decrease of 13.22% in susceptible plants. Tyrosine was increased by 45.03% only in the tolerant plants when applied with PGPR, while it was decreased by 51.52% in susceptible plants. Similarly, lysine was increased (35.12%) in the PGPR-applied tolerant plants, while no significant difference was observed in the susceptible plants. In contrast, proline was increased by 35.77% only in susceptible plants, while no significant difference was recorded in intolerant plants when challenged with PGPR (Table 2).

## DISCUSSION

The use of microbial-based techniques in the management of plant diseases has gained significant attention in recent years. In particular, PGPRs and their interactions with the plants under biotic or abiotic stress are gaining importance, with the ultimate aim of improvement in the protection of crops and increases in agricultural production. These biocontrol approaches are eco-friendly and are becoming very popular, reliable, and long-lasting. Plant growth improvement by PGPR is one of the outstanding characteristics of these naturally occurring microbes. The improvements in plant growth and its ameliorating abilities about plant diseases are determined by the interactions between the host plant and PGPR (*Vejan et al., 2016*). PGPR improves plant growth and health by direct or indirect mechanisms that can overcome diseases. The plant growth-promoting activity of PGPR bacteria has been reviewed in detail by *Santoyo et al. (2016)* (*Xia et al., 2015*; *Santoyo et al., 2016*). *Bacillus* and *Pseudomonas* species are widely known as invaluable resources for plant growth promotion and the suppression of disease symptoms (*Sundaramoorthy & and Balabaskar, 2013*; *Chaves-López et al., 2015*). Over the last few decades, several studies have reported on the beneficial aspects of *Bacillus* spp. as biocontrol and biofertilizer agents; e.g., *Bacillus licheniformis*, *Bacillus subtilis*, *Bacillus cereus*, *Bacillus pumilus*, and *Bacillus amyloliquefaciens* (*Pane & Zaccardelli, 2015*; *Han et al., 2016*). The plant growth promotion and other beneficial aspects of *Bacillus* strains can be attributed to their ability to enhance the production of phytohormones such as auxin (IAA), and gibberellic acid (*Gamalero & Glick, 2011*).

A wide range of plant species is infected by pathogens, including the diverse genera of *Alternaria*, *Botrytis*, *Fusarium*, and *Rhizoctonia*. These pathogens result in severe losses to

**Table 2   Regulation of amino acids (mg/g DW) in tomato plants applied withSBR02 in the presence and absence of *F. oxysporumf. sp.lycopersici*.**

| Cultivar | Treatment | Asp | Thr | Met | ILE | Ser | Glu | Leu | Tyr | Gly | Phe | Lys | Cys | Val | His | Arg | Ala | Pro |
|---|---|---|---|---|---|---|---|---|---|---|---|---|---|---|---|---|---|---|
| | Control | 4.93 ± 0.12b | 4.84 ± 0.27b | 0.69 ± 0.17b | 4.95 ± 0.22b | 4.83 ± 0.11b | 12.5 ± 0.32b | 9.91 ± 0.11b | 3.30 ± 0.60a | 5.73 ± 0.22b | 3.38 ± 0.32b | 6.63 ± 2.69b | 0.36 ± 0.02a | 6.10 ± 0.21b | 2.67 ± 0.32b | 4.49 ± 0.45b | 6.31 ± 0.21b | 9.87 ± 1.15b |
| | SRB02 | 6.16 ± 0.07a | 4.71 ± 0.27a | 0.91 ± 0.08a | 6.41 ± 0.24a | 5.30 ± 0.14a | 16.1 ± 0.33a | 13.13 ± 0.4a | 1.60 ± 0.15b | 5.80 ± 0.16a | 5.99 ± 0.14a | 7.73 ± 1.52a | 0.57 ± 0.03a | 8.05 ± 0.14a | 3.78 ± 0.31a | 5.56 ± 0.47a | 6.60 ± 0.18a | 13.4 ± 0.74a |
| Tolerant | *F. oxy* | 2.96 ± 0.12b | 3.05 ± 0.27b | 0.32 ± 0.06b | 2.51 ± 0.44b | 2.70 ± 0.14b | 7.36 ± 0.35b | 6.21 ± 0.20b | 2.29 ± 0.2b | 3.24 ± 0.14b | 5.92 ± 0.18b | 5.29 ± 0.62b | 0.53 ± 0.05a | 3.54 ± 0.42b | 1.25 ± 0.13b | 2.82 ± 0.3b | 4.06 ± 0.06b | 4.67 ± 0.56b |
| | SRB02 + *F. oxy* | 7.33 ± 0.46a | 6.66 ± 0.22a | 0.88 ± 0.02a | 4.98 ± 0.09a | 7.01 ± 0.22a | 18.99 ± 0.2a | 9.82 ± 0.26a | 3.79 ± 0.25a | 7.90 ± 0.15a | 7.94 ± 0.19a | 10.49 ± 2.26a | 0.41 ± 0.01b | 6.18 ± 0.19a | 2.66 ± 0.33a | 7.26 ± 0.6a | 8.52 ± 0.34a | 10.0 ± 0.7a |
| | Control | 6.14 ± 0.07b | 4.41 ± 0.38b | 0.73 ± 0.1b | 3.02 ± 0.14b | 5.90 ± 0.09b | 7.77 ± 0.8b | 9.28 ± 0.24b | 2.50 ± 0.30a | 5.31 ± 0.26b | 3.57 ± 0.24b | 8.44 ± 0.38b | 0.39 ± 0.04a | 6.30 ± 0.36b | 2.96 ± 0.09b | 4.67 ± 0.32b | 6.36 ± 0.43b | 11.5 ± 1.09b |
| | *SRB02* | 6.34 ± 0.13a | 5.87 ± 0.12a | 1.71 ± 0.38a | 5.99 ± 0.14a | 5.12 ± 0.11a | 14.1 ± 0.69a | 12.67 ± 0.54a | 3.60 ± 0.20b | 6.95 ± 0.1a | 7.69 ± 0.36a | 11.4 ± 1.08a | 0.55 ± 0.01a | 7.68 ± 0.29a | 3.63 ± 0.44a | 6.35 ± 0.17a | 8.25 ± 0.22a | 12.5 ± 0.56a |
| Susceptible | *F. oxy* | 3.45 ± 0.47b | 2.94 ± 0.13b | 0.43 ± 0.05b | 4.58 ± 0.35b | 2.90 ± 0.13b | 16.87 ± 0.15b | 6.19 ± 0.30b | 1.60 ± 0.30a | 3.28 ± 0.2b | 5.60 ± 0.29b | 5.33 ± 0.36b | 0.69 ± 0.07a | 3.95 ± 0.16b | 1.22 ± 0.28b | 3.15 ± 0.23b | 4.20 ± 0.21b | 5.55 ± 0.38b |
| | SRB02 + *F. oxy* | 7.45 ± 0.23a | 6.57 ± 0.47a | 1.47 ± 0.20a | 6.54 ± 0.44a | 6.48 ± 0.38a | 17.69 ± 0.22a | 12.59 ± 0.38a | 3.90 ± 0.5a | 7.60 ± 0.37a | 7.83 ± 0.27a | 11.37 ± 0.64a | 0.62 ± 0.04b | 7.83 ± 0.19a | 2.43 ± 0.17a | 6.57 ± 0.39a | 8.38 ± 0.25a | 10.55 ± 0.38a |

**Notes.**

Asp, Aspartic acid; Thr, Threonine; Met, Methionine; ILE, Isoleucine; Ser, Serine; Glu, Glutamic acid; Leu, Leucine; Tyr, Tyrosine; Gly, Glycine; Phe, Phenylalanine; Lys, Lysine; Cys, Cysteine; Val, Valine; His, Histidine; Arg, Arginine; Ala, Alanine; Pro, Proline.

Each value represents means ± SD of three independent experiments. Each values in columns followed by different letters are significantly different at $P \geq 0.05$.

crop yield and productivity, thereby posing a threat to food security. *F. oxysporum* is a devastating fungal pathogen that attacks the vascular system and causes severe damages to tomato crops across the globe. Conversely, microbes, or PGPRs, found in the rhizosphere of plants are directly associated with roots and are a vital source for plant growth promotion and suppression of soilborne plant pathogens such as *F. oxysporum*. To isolate and evaluate the beneficial role of PGPR, an appropriate in vitro experimental setup is required. *Shahzad et al. (2017a)*, *Shahzad et al. (2017b)* reported plant growth promotion by endophytic bacteria RWL-1 against the pathogenic infection by *FOL* in tomato. Also, it was recently reported that *B. aryabhattai SRB02* plays a role in oxidative and nitrosative stress tolerance and promotes the growth of soybean and rice plants by modulating the production of phytohormones (*Park et al., 2017a*). However, it was not clear whether *B. aryabhattai SRB02* could be used to rescue the plants from biotic stress. Hence, in the present study, we subjected disease-tolerant and susceptible tomato plants to the PGPR *B. aryabhattai* SRB02 in the presence and absence of a virulent strain of *FOL,* hypothesizing that SRB02 would rescue the plants from the disease and improve their growth under stress conditions. Before inoculation by the pathogen, tomato plants were treated with a cell suspension of *B. aryabhattai* SRB02. The SRB02 application improved the disease tolerance level of the infected plants. In a previous study by *Shahzad et al. (2017a)*, *Shahzad et al. (2017b)*, PGPRs were shown to enhance plant growth, reduce infection by the pathogen, and result in improved disease tolerance.

The present study showed that under pathogenic infection, the PGPR association rescued the plants from disease and enhanced plant growth and biomass. This result might occur by restricting the pathogenic fungus, enhancing nutrient uptake, and producing phosphate solubilization substances, or by induction of phytohormonal biosynthesis. The present findings further strengthen the role of *Bacillus* species as a PGPR and biocontrol agent, as reported by numerous researchers, against diverse diseases in various plant species, such as root wilting, damping off, fusarium wilt, ring rot, and charcoal rot in tomato, soybean, banana, apple, and common bean, respectively (*Yu et al., 2002*; *Vitullo et al., 2012*; *Wang & Fobert, 2013*; *Chen et al., 2016*; *Torres et al., 2016*). The current findings also indicate that PGPR strains producing bioactive components may suppress the negative effects of pathogenesis and biotic stress in infected plants. In addition, our study also confirmed and exhibited similar results to previous reports that organic acids, as one of the many components produced by *Bacillus* species, can help rescue the plant from the disease. Moreover, PGPR produces siderophores and organic acids, which mitigate the negative effects of pathogen-infected sunflower plants (*Waqas et al., 2015*). From these studies, it is evident that biotic stress-related ameliorative effects are commonly regulated by endogenous phytohormones such as SA and JA. Under normal and stress conditions, phytohormone signaling and crosstalk play a vital role in plant growth and development. Accordingly, in the present study, we found that inoculation with PGPR *B. aryabhattai SRB02* extensively modulated the endogenous levels of JA and SA. Our findings conform with previously elucidated phytohormonal regulation; i.e., increased SA (Fig. 4) and reduced JA (Fig. 3) with PGPR, as revealed by independent studies (*Khan et al., 2015*;

*Waqas et al., 2015*; *Shahzad et al., 2016*; *Shahzad et al., 2017b*; *Ali et al., 2017*) comparing plants in the presence or absence of biotic stress.

### Funding
This work was supported by a grant from the Next-Generation BioGreen 21 Program Rural Development Administration, (SSAC, Grant NO. PJ01242501), Republic of Korea. The funders had no role in study design, data collection and analysis, decision to publish, or preparation of the manuscript.

### Grant Disclosures
The following grant information was disclosed by the authors:
Next-Generation BioGreen 21 Program Rural Development Administration: PJ01242501.

### Competing Interests
The authors declare there are no competing interests.

### Author Contributions
- Rizwana begum Syed Nabi conceived and designed the experiments, performed the experiments, prepared figures and/or tables, authored or reviewed drafts of the paper, and approved the final draft.
- Raheem Shahzad conceived and designed the experiments, performed the experiments, analyzed the data, authored or reviewed drafts of the paper, and approved the final draft.
- Rupesh Tayade analyzed the data, prepared figures and/or tables, authored or reviewed drafts of the paper, and approved the final draft.
- Muhammad Shahid performed the experiments, analyzed the data, prepared figures and/or tables, and approved the final draft.
- Adil Hussain and Byung-Wook Yun conceived and designed the experiments, authored or reviewed drafts of the paper, and approved the final draft.
- Muhammad Waqas Ali analyzed the data, prepared figures and/or tables, and approved the final draft.

### Data Availability
Raw data are available in the Supplementary Files.

### Supplemental Information
Supplemental information for this article can be found online at http://dx.doi.org/10.7717/peerj.11194#supplemental-information.

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
