# Peer review of "Evaluation potential of PGPR to protect tomato against Fusarium wilt and promote plant growth"

_PeerJ, doi:10.7717/peerj.11194_

## Round 0.1 · original submission · Major Revisions

Dear Dr.Syed Nabi,

Your manuscript, which you submitted to PeerJ, has been reviewed by two experts in your research area. The comments from reviewers are included at the bottom of this letter and in the attachments.

Both reviewers recommended MAJOR REVISIONS and pointed out that the manuscript needs significant improvement. I found the meanings of many sentences are hard to capture. Therefore, I encourage you to get with an English-speaking professional person to edit your revision to avoid some jargon and more clear.

My review copy also attached for your information.

Thank you again for submitting your manuscript to PeerJ for publication.

Best regards,

Sincerely,

Tika Adhikari

Reviewer 1 ·

Basic reporting

no comment

Experimental design

no comment

Validity of the findings

no comment

Additional comments

The manuscript by Rizwana begum Syed Nabi et al. evaluated the ability of plant growth-promoting rhizobacterium (PGPR) Bacillus aryabhattai strain SRB02 to alleviate the effects of tomato wilt disease caused by Fusarium oxysporum f. sp. lycopersici (strain KACC40032) and promote plant growth, which is very beneficial to protect crops from vascular wilt, one of the most important fungal diseases of crops.

Major concerns:
1. Lack of the research data on the mechanism of bacteria inhibiting fungi;
2. Figure 1 has poor clarity and the name of the bacteria and fungus in the plate was not indicated;
3. Figs 4 and 5, the meaning of the significant difference mark are not stated in the legend. The difference between tolerant or susceptible plants should be marked with different signs;
4. Table 1 and Table 2, Table names and Tables are mismatch; data with significant differences between different treatments should be marked with symbols;
5. Line 88 and 279, pointed out that one of the mechanisms of PGPR to promote plant growth is that PGPR synthesizes ethylene, which is incorrect. Instead, PGPR degrades the ACC synthesized by plants through ACC deaminase, thereby reducing the plant ethylene synthesis and the inhibition of ethylene on plants.

Minor concerns:
Line 325, 343, 346, 355, 371, 383, 387, 404, 418, 433, the reference format is incorrect.

Reviewer 2 ·

Basic reporting

The research work described the ability of plant growth-promoting rhizobacterium Bacillus aryabhattai strain SRB02 to control tomato wilt disease caused by Fusarium oxysporum f. sp. Lycopersici. The paper presents an interesting research topic and good ideas, but the manuscript is not acceptable for publication in its present form. The research work contains insufficient methodological detail. The plant growth-promoting activity and bio-control of fungal disease of tomato by B. aryabhattai were reported (see comments)., thus it should be indicated and cited. Most of references are outdated and should be replaced by recent reports. Tables and figures should be worked out and statistical analyses should be added.

Experimental design

The research work contains insufficient methodological detail. I have made some comments that should be improved and more detailed descriptions should be provided. Research questions should be also worked out because PGP traits of used bacteria were not described in detail.

Validity of the findings

no comment

Additional comments

Comments
Introduction section: Please reduce outdated citations and replace them with recent reports. The introduction part should focus on the biological control of plant pathogens by PGPR and, mechanisms of action.
Line 23: “to alleviate the effects of tomato wilt disease caused” to alleviate or to control?
Line 96-98: add citations.
Line 111: “there is a lack of information about the growth-promoting activity of B. aryabhattai”. There are some reports on the effect of this bacteria on plant growth of tomato, and biological control, please add references. https://pubmed.ncbi.nlm.nih.gov/31216607/
https://link.springer.com/article/10.1007/s40858-019-00283-2

Line 132:” tomato seeds of four Korean cultivars” include cultivar names and describe each cultivar characterization.
Line 134: Horticultural soil, describe soil physical and chemical characteristics and origin
Line 136: big pots, the volume?
Line 142: 14 days growth, is the plant growth duration not so short to show symptoms of the disease?
Line 146: biotic stress? Please describe
Line 153: why plants were grown in high soil humidity? Plant growth conditions should be described in detail.
Line 155: “. to determine the response of plants to infection in the presence or absence of SRB02”, dry weigh or chlorophyll content does not relate to plant-pathogen interaction, other parameters should be discussed to support the response of the plant to fungal infection.
Line 164:” was determined using relevant standards.” Describe in detail
Line 199: Response of tomato cultivars against pathogen?
Figure 1, please add as supplementary material

Figure 2. This figure is not clear, clarify control and infected plant, and also describe means of tolerant, moderately tolerant and susceptible varieties. How these parameters were determined?
Figure 6 demonstrate general mechanisms used by PGPR to control plant disease and stimulate plant growth, but not findings of this study, thus should be omitted.
Table 2 has some format problems, statistical analyses are missing
Table 1. Statistical analyses are missing, add significant differences among treatments

---

## Round 0.2 · accepted · Accept

Dear Dr. Syed Nabi,

Thank you for your submission to PeerJ. I am writing to inform you that your manuscript - Evaluation potential of PGPR to protect tomato against Fusarium wilt and promote plant growth - has been Accepted for publication.

Congratulations!

Tika Adhikari